# Experimental Investigation of Surface Roughness in Milling of Duralcan^TM^ Composite

**DOI:** 10.3390/ma14206010

**Published:** 2021-10-12

**Authors:** Martyna Wiciak-Pikuła, Paweł Twardowski, Aneta Bartkowska, Agata Felusiak-Czyryca

**Affiliations:** 1Faculty of Mechanical Engineering, Institute of Mechanical Technology Poznan, University of Technology, 3 Piotrowo St., 60-965 Poznan, Poland; pawel.twardowski@put.poznan.pl (P.T.); agata.z.felusiak@doctorate.put.poznan.pl (A.F.-C.); 2Faculty of Materials Engineering and Technical Physics, Institute of Materials Science and Engineering, University of Technology, 3 Piotrowo St., 60-965 Poznan, Poland; aneta.bartkowska@put.poznan.pl

**Keywords:** metal matrix composites, surface roughness, CART, ANN

## Abstract

In today’s developing aircraft and automotive industry, extremely durable and wear-resistant materials, especially in high temperatures, are applied. Due to this practical approach, conventional materials have been superseded by composite materials. In recent years, the application of metal matrix composites has become evident in industry 4.0. A study has been performed to analyze the surface roughness of aluminum matrix composites named Duralcan^®^ during end milling. Two roughness surface parameters have been selected: arithmetical mean roughness value *Ra* and mean roughness depth *Rz* regarding the variable cutting speed. Due to the classification of aluminum matrix composites as hard-to-cut materials concerning excessive tool wear, this paper describes the possibility of surface roughness prediction using machine learning algorithms. In order to find the best algorithm, Classification and Regression Tree (CART) and pattern recognition models based on artificial neural networks (ANN) have been compared. By following the obtained models, the experiment shows the effectiveness of roughness prediction based on verification models. Based on experimental research, the authors obtained the coefficient R^2^ for the CART model 0.91 and the mean square error for the model ANN 0.11.

## 1. Introduction

Nowadays, composites materials are used extensively by the automotive and aircraft industries because of their specific mechanical properties. One of the popular construction materials is metal matrix composites (MMCs), which offer higher hardness and wear resistance than conventional monolithic materials. These multiphase materials containing matrix and reinforcement are characterized by specific strength and good wear resistance. Moreover, these reinforcing phases efficiently increase the modulus due to MMCs’ load-bearing capability during mechanical loading [1,2]. One type of MMCs is particles reinforced metal matrix composites (PRMMCs), including various kinds of particles added into the metal matrices such as carbides, oxides, or nitrides. Compared to the other form of reinforcement such as whiskers, short fibers, and continuous fiber, the particles give better isentropic properties to distribute uniformly in the matrix phase. In addition, the reinforcement in the form of particles bears a higher load than the matrix, which strengthens it effectively (called the load transfer effect). Unfortunately, if the load on the particle strength is exceeded, the particle will crack. Therefore, PRMMCs are exposed to early fracture, and strength and ductility reduction. Thus, during the design and fabrication of composites with particles reinforcement, it is essential to consider particle size, aspect ratio, and matrix strength to reduce particle damage [3].

One of the most used industrial applications is aluminum matrix composites (AMCs), which consist of pure aluminum or aluminum alloy as a matrix. Aluminum alloys are one of the most common nonferrous metals used in commercial production, and these alloys, with reinforcement, can offer outstanding mechanical and tribological properties. The principal advantages of reinforcement in AMCs are the improvements in hardness, tensile strength, impact strength, compressive strength, and wear resistance [4]. These types of composites are produced by in situ fabrication, rheocasting, and spray deposition in semi-solid matrix conditions, but the most popular is solid- and liquid-state processing [5]. The most commercially used technique in liquid-state processing is stir casting, which is more cost-effective than solid-state methods. This method provides a relatively homogenous dispersion of reinforcements in a matrix [6]. One typical reinforcement used in the aviation and manufacturing industry is silicon carbide (SiC) owing to its thermal characteristics and tribological properties. S. Sivananthan et al. [7] successfully adopted stir casting to fabricate AA 6061 alloy with 0–4 wt.% of SiC particulates. The SiC composite acquired better properties than AA 6061 alloy, and just four wt.% of SiC improved hardness by 25% and tensile strength by 25.6% compared to AA 6061 alloy. The enhancement of mechanical properties was also observed by J. Jebeen Moses et al. [8], who produced AMC with SiC particulates of 5, 10, and 15 wt.%. They reported that AA 6061 alloy with 15% volume fraction improved microhardness by 133.33% and ultimate shear by 65.2% compared to AA 6061 alloy. Moreover, such an approach allows homogenously distributing SiC particulates in the aluminum alloy matrix and limiting the risk of SiC segregation along the grain boundaries.

Nevertheless, aluminum-based silicon carbide composite (SiCp/Al) is also a hard-to-cut material because of its high-hardness reinforcing particles, which could cause excessive tool wear and deterioration of surface roughness. For high precision engineering applications, material removal rate, tool life, and workpiece surface roughness are necessary for machinability assessment. Nowadays, most researchers focus on the influence of various SiCp/Al composite parameters on its machinability. For example, P. Zhang et al. [9] analyzed the size particle effect on cutting force, cutting temperature, and chip shape under different cutting parameters. The results show that cutting force is positively correlated to feed rate and particle size. When the particles are between 10 and 30 μm and the cutting speed increases, the main cutting force decreases. On the other hand, when particle size increases to 40 μm, the cutting force firstly decreases and then increases. Moreover, the bending radius and length of the chip decrease as particle size increases. Other studies are focused on the machining condition effect on surface roughness, especially examining the influence of the cutting parameters [10,11]. Attention is focused on the surface roughness to obtain a good fatigue life of machined parts. PRMMCs’ surface roughness is complex because of voids, microcracks, pits, protuberances, grooves, or matrix tearing on the machined surface. Additionally, the irregular surface texture is observed due to particles in the matrix, and particle fracture affects the surface roughness [12,13,14]. Khare et al. [15] investigated the influence of cutting parameters and wt.% Al_2_O_3_/Gr on surface roughness during the end milling of Al/Al_2_O_3_/Gr composite. This study shows a significant feed rate, cutting speed, and wt.% Al_2_O_3_/Gr on surface roughness. In addition to cutting parameter influence, the optimization of machining conditions in the machining of MMCs is general in today’s studies [16,17]. The optimization of machining conditions in MMCs milling was studied by S. Karabulut et al. [18] to achieve a better surface finish. In this study, the machining parameter was optimized by Taguchi’s L18 (21 × 32), and an artificial neural network (ANN) model was used to estimate Ra’s arithmetical mean roughness value. Results show the best optimal parametric combination for AA7039/SiC and AA7039/B_4_C milling (cutting speed 488 m/min and feed rate 0.1 mm/tooth) and the most significant cutting speed impact on surface finish. Due to the poor machinability of MMCs and high processing costs, soft computing techniques and unconventional machining [19] have become a significant interest for researchers to determine performance prediction and optimization. Some of the soft techniques applied by researchers due to analyzing the machining of MMCs are ANN, response surface methodology (RSM), genetic algorithm (GA), Taguchi method, or finite computational element [20,21]. These techniques are relatively cost-effective and could be applied in numerical and experimental approaches for modeling or simulating MMCs’ manufacturing and machining [22]. A numerical model for analyzing the effects of PMMCs particulate size on damage mechanisms was developed by S. Gad et al. [23]. This paper’s computational finite element (FE) model was proposed to determine the elastoplastic behavior of A359/SiC particulate composite. They concluded that increasing SiC volume fracture (from 2 to 20%) leads to the increased modulus of elasticity, yield strength, and tensile strength. In addition, raising the particulate size in the matrix reduces the yield strength, ultimate tensile strength, and failure strain. These kinds of models could be helpful in the optimization of reinforcement parameters during the design phase. One of the new approaches to support decision making in MMCs casting was proposed by R. Sika et al. [24]. They designed Open Atlas of Casting Defects (OACD) to identify various defects of casting. Such a classification of defects in MMCs castings could be an appropriate tool to eliminate these defects using a support expert program. Soft techniques are also used to optimize surface roughness [25,26]. S. Karabulut [27] applied the Taguchi method to optimize cutting parameters, and ANN to predict surface roughness during milling of AA7039/Al_2_O_3_ metal matrix composites. Their experiments showed that the best cutting parameters for superior surface roughness were observed for a cutting speed of 488 m/min, feed rate of 0.1 mm/tooth, and axial depth of cut of 1 mm. They also developed an effective ANN prediction model for surface roughness achieving determination coefficient R^2^ = 97.75%. G. Zhou et al. [28] proposed an ANN roughness prediction model for Al/SiC particulate composite material milling. They developed a learning method to solve the MMCs milling problems effectively, and a successfully trained ANN model that could predict surface roughness with a 2.08% mean relative error.

Most past studies focus on optimizing cutting parameters and analyzing cutting forces, surface roughness, and tool wear in MMCs machining. Optimization and prediction techniques are mostly used by the Taguchi method, ANN, or analysis of variance (ANOVA) to machining aluminum particulate composites. The scope of this paper involves the application of Classification and Regression Tree (CART) and pattern recognition models based on ANN to predict surface roughness in Al/SiC particulate composites with 10% volume fracture. Adaptation of these kinds of soft techniques aims to understand the machinability problem of hard-to-cut composites. In addition, analyzing surface roughness could be valuable for future researchers.

## 2. Materials and Methods

The end milling investigation was carried out using SiC particle-reinforced aluminum alloy composites called Duralcan^TM^. This material is manufactured by mixing the ceramic powder into molten aluminum, using a patented process. Then, the melt is poured into the foundry ingot, and products are formed using high-pressure die-casting. In this paper, the F3S.10S (AA359/SiC/10p) was applied to experimental studies. The range of mechanical and physical properties are shown in Table 1; Table 2. These composites have many uses in manufacturing in the automotive industry, such as brake rotors, brake calipers, brackets, and brake drums, etc. 

The scanning electron microscopy (SEM) integrated with energy dispersive spectroscopy (TESCAN MIRA3 FEG SEM, Brno, Czech Republic) was applied to evaluate the morphology of AA359/SiC/10p composite. The metallographic microsections end EDM micrographs of AMC are shown in Figure 1.

The dry end milling experiments were conducted on a DECKEL-MAHO DMC70 V (Pfronten, Bayern, Germany) machining center integrated with a piezoelectric force sensor. Furthermore, diamond-coated end mills were chosen to carry out research (diameter of cutting-edge *d* = 10 mm, number of edges *z* = 3). In Table 3, the research plan with one variable is presented. 

Three repetitions for each cutting speed were carried out. After each five milling pass, the cutting force components were measured in three directions (*Fp* for the axial direction, *FfN* for normal feed direction, and *Ff* for feed direction). One of the stages of the research was also the measurement of roughness parameters *Ra* (arithmetic mean roughness) and *Rz* (surface roughness depth), and tool corner wear *VB_C_.* For this purpose, the Hommel Tester T500 (JENOPTIK Industrial Metrology, Villingen-Schwenningen, Germany) profilometer measured length *ln* = 4 mm, and the elementary segment *lr* = 0.08 mm was applied to assess the topography of the machined surface. Figure 2 shows the scheme of the experimental end milling process. 

## 3. Results 

### 3.1. Analysis of Surface Roughness

The analysis of *Ra* and *Rz* parameters was investigated in various cutting speeds to determine the relations between the surface roughness and cutting forces. MMC machined surfaces at different cutting speeds are given in Figure 3, Figure 4 and Figure 5.

To present the measured value of surface roughness during the end milling of Duralcan^TM^, the *Ra* and *Rz* parameters in cutting time function are shown in Figure 6 and Figure 7. The uneven rise of surface roughness could be caused by the unbalanced distribution of SiC particles in the matrix. As a result, the SiC particles are tearing from the matrix, and the machined surface quality is not satisfactory. Moreover, excessive tool wear causes microcracks and pits on the AMC surface. The relationship between the surface roughness, cutting speed, and tool corner wear (*VB_C_*) is presented in Figure 8.

### 3.2. Analysis of Cutting Forces

Analysis of the cutting force’s components in the time domain and the frequency domain was conducted to recognize the correlation between surface roughness parameters and measured signals. During the tests, root mean square values (RMS) based on the time domain were selected. Additionally, tool revolution frequency (*Ffr*) was identified in the frequency domain. The tool’s revolution frequency was calculated for three cutting speeds
(1)$$fr=\frac{n}{60}\cdot z\;$$
where *n* is the spindle speed (rev/min) and *z* is the number of edges.

An exemplary relation between the *Ra* and diagnostic measures at various cutting speed is presented in Figure 9. 

The R^2^ coefficient determines the matching of mathematical function to the results of the test. 

Figure 10 presents an exemplary correlation between the selected cutting components (diagnostic measures) and surface roughness. The low coefficient R^2^ indicates the possibility of using more complex models than regression. 

### 3.3. Diagnostic Model Based on Classification and Regression Tree (CART)

One of the classifications and predictive methods is CART, which creates the possibility of representing the knowledge after the learning process. This kind of method is easy to implement in the diagnostic procedures to develop an independent expert system. In this paper, the CART and Chi-squared Automatic Interaction Detector (CHAID) Tree was proposed to predict the surface roughness based on cutting forces. The structure of the CHAID tree with three interior nodes and four final nodes for *Ra* prediction is presented in Figure 11.

The CHAID method consists of trees where each node contains a split condition, and its purpose is optimal prediction, especially in regression problems. The cutting speed, cutting forces, and tool wear were entered as inputs. Based on the selected input data, the validity analysis was carried out. The study shows that the *FfN_RMS_* diagnostic measure has the most significant impact on surface roughness *Ra* (Figure 12). The data set was divided into two subsets: training and testing. The test subset aims to assess the model’s generalizability and accounts for 30% of the input data. To check the effectiveness of the predicted model, the measured and indicated value was compared. A spread graph of this value is presented in Figure 13.

Different types of models were carried out to select the best predictive model with the most significant efficiency. Another analysis was developed using CART with four interior nodes and five final nodes. The model consists of many simple models built on subsamples drawn from the training set for CART trees. Earlier, the case weights were determined that increased the probability of being drawn to the next set of these cases that generated the most significant error. The structure of CART is presented in Figure 14. 

The effectiveness of the prediction model was checked, similarly to CHAID analysis, and the comparison of the predicted and observed values of *Ra* is shown in Figure 15. The mean square error (MSE) was estimated to compare measured and expected surface roughness based on the new experimental data. For this CART model, MSE for the verification model is 0.026. 

### 3.4. Diagnostic Model Based on Artificial Neural Network (ANN)

With a view to select the best surface roughness predictive model, a Multilayer Perceptron (MLP) was also developed. For this analysis, the number of random samples was assumed at 70% for the training set, 15% for the test, and 15% for the validation set. The input data were cutting force components, cutting speed, and tool wear, the same as in CART models. In Table 4, the characteristic parameters of the MLP model are shown. In this model, ten hidden layers were assumed. In this case, the validity analysis was performed and showed that the *Fp_RMS_* diagnostic measure has the most significant impact on surface roughness *Ra* (Figure 16). Based on the new experimental data, the MSE is 0.11, and the comparison of predicted and observed values of *Ra* is shown in Figure 17.

## 4. Conclusions

The results of surface roughness studies on AA359/SiC/10p composites during end milling were presented. The possibility of implementing the computing techniques to predict the surface roughness parameter during the machining of hard-to-cut composites are observed, and the following conclusions have been drawn: The prediction of surface roughness based on the cutting forces is conceivable. Still, it is necessary to implement another type of model rather than regression because of the low determination coefficients (R^2^*_Ra_* = 0.67, and R^2^*_Rz_* = 0.32) due to excessive tool wear and pits on the Duralcan™ surface.The application of ANNs to predict surface roughness gives a satisfactory effect and the possibility to achieve a diagnostic system based on cutting force’s measures. The mean square error for the verification model is 0.11.The decision tree method is a basic predictive model, which might be achieved in milling metal matrix composites. The applied CART model gives better results than MLP, whereby the best effect was observed for the CART verification model (R^2^ = 0.91).In summary, computing techniques such as machine learning or artificial intelligence are straightforward methods that could be used to predict surface roughness during the machining of particle-reinforced aluminum alloy composites.

## Figures and Tables

**Figure 1 materials-14-06010-f001:**
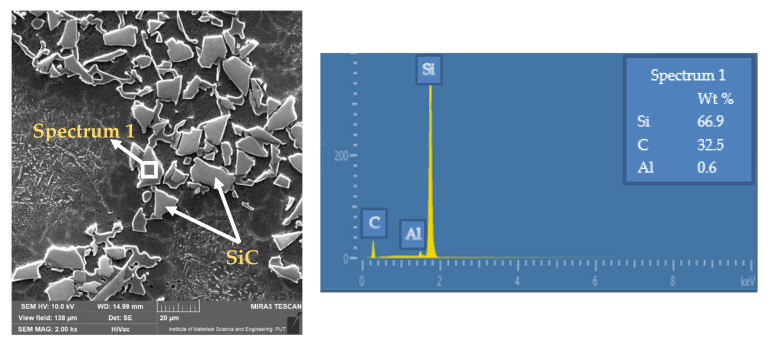
Metallographic microsections of F3S.10S composite and EDS micrographs of SiC powder.

**Figure 2 materials-14-06010-f002:**
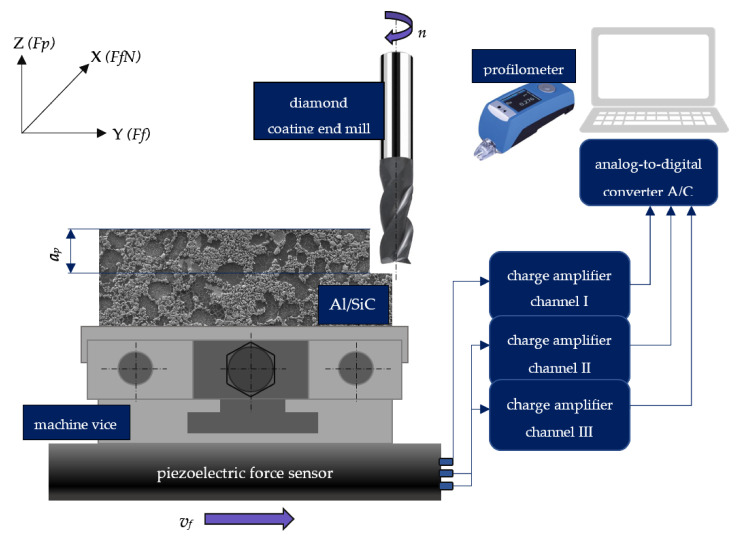
Schematic of the experimental set up.

**Figure 3 materials-14-06010-f003:**
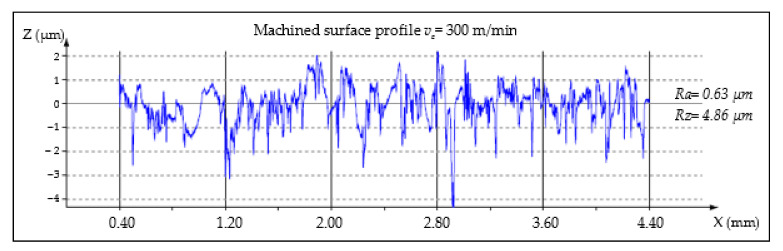
Machined surface profile of the AA359/SiC/10p, *v_c_* = 300 m/min, *t_c_* = 3.65 min.

**Figure 4 materials-14-06010-f004:**
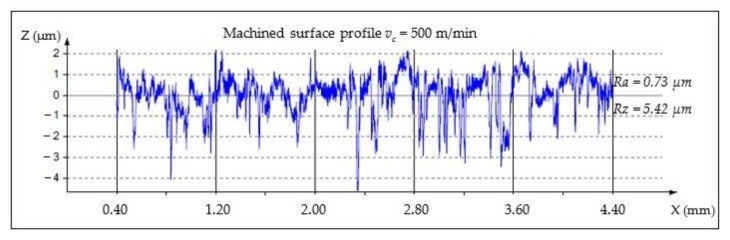
Machined surface profile of the AA359/SiC/10p, *v_c_* = 500 m/min, *t_c_* = 1.82 min.

**Figure 5 materials-14-06010-f005:**
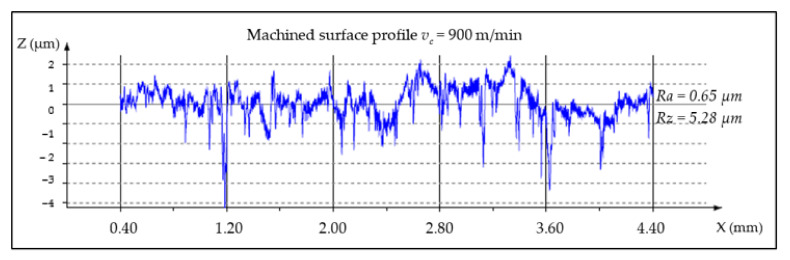
Machined surface profile of the AA359/SiC/10p, *v_c_* = 900 m/min, *t_c_* = 1.22 min.

**Figure 6 materials-14-06010-f006:**
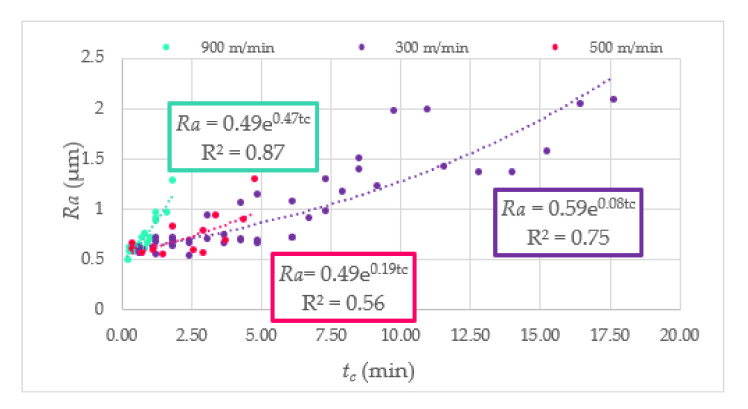
The *Ra* parameter in function of cutting time.

**Figure 7 materials-14-06010-f007:**
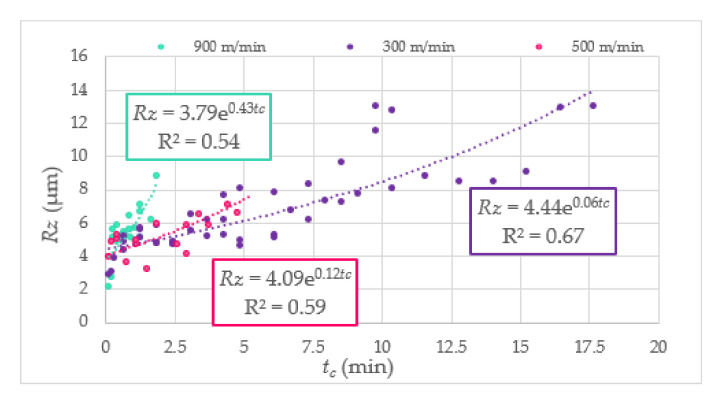
The *Rz* parameter in function of cutting time.

**Figure 8 materials-14-06010-f008:**
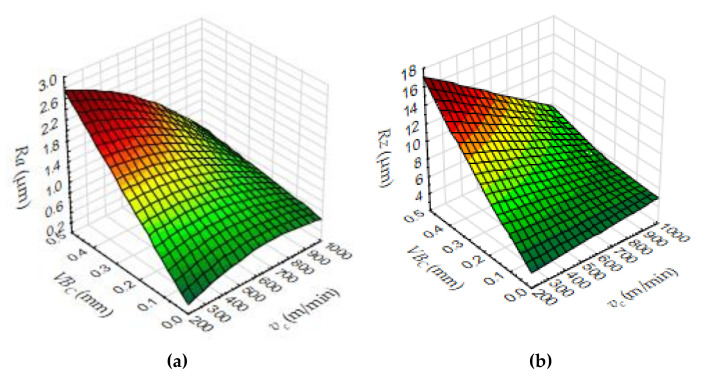
Three dimensional surface plot, (**a**) *Ra* dependence on *VB_C_* and *v_c_*, (**b**) *Rz* dependence on *VB_C_* and *v_c_*.

**Figure 9 materials-14-06010-f009:**
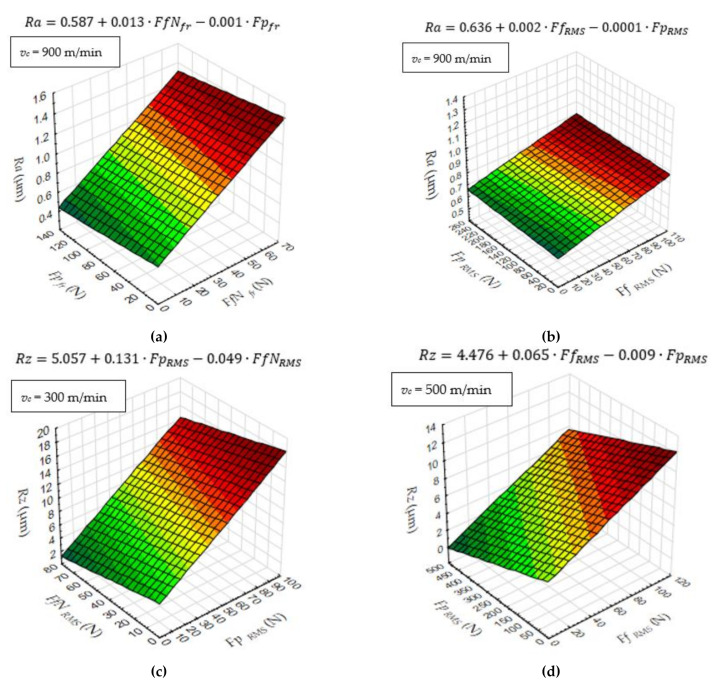
Three dimensional surface plot, (**a**) *Ra* dependence on *Fp_fr_* and *FfN_fr_*, (**b**) *Ra* dependence on *Fp_RMS_* and *Ff_RMS_*, (**c**) *Rz* dependence on *FfN_RMS_* and *Fp_RMS_*, (**d**) *Rz* dependence on *Fp_RMS_* and *Ff_RMS_*.

**Figure 10 materials-14-06010-f010:**
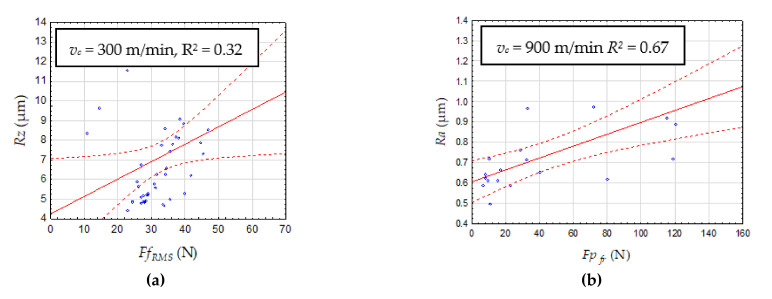
Surface roughness parameter as a function of cutting force measures (**a**) time domain, *Ff_RMS_*, (**b**) frequency domain *Fp_fr_*.

**Figure 11 materials-14-06010-f011:**
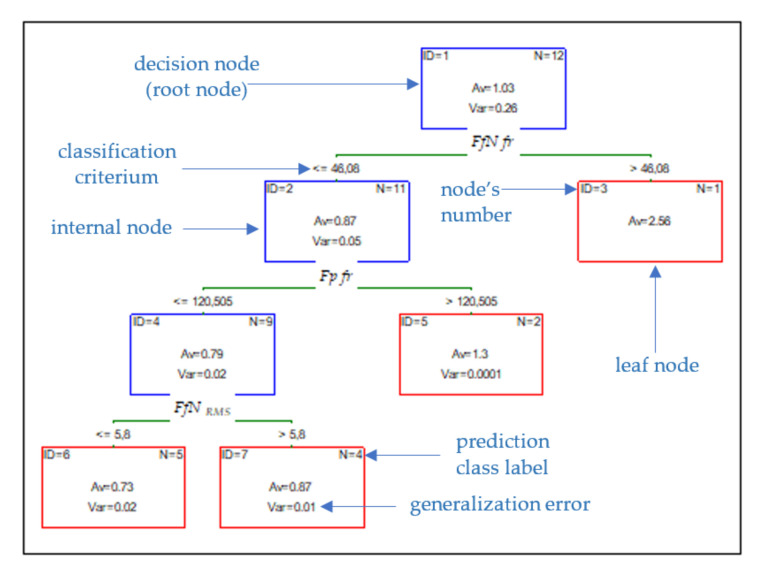
Structure of CHAID tree for parameter *Ra*.

**Figure 12 materials-14-06010-f012:**
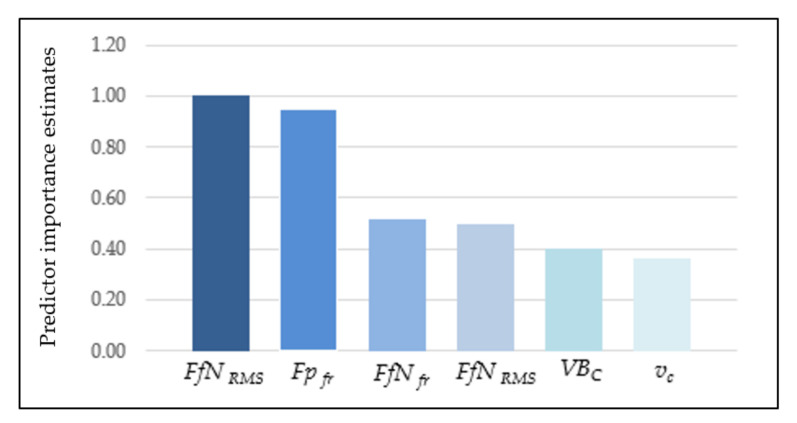
Validity of input parameters on surface roughness (CHAID).

**Figure 13 materials-14-06010-f013:**
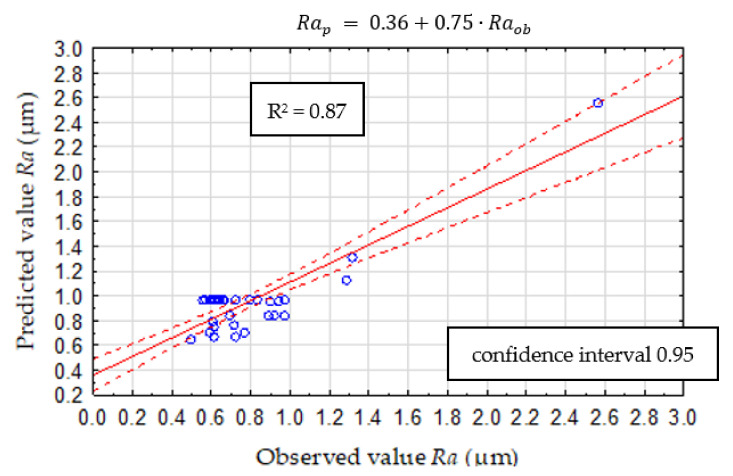
CHAID validation model.

**Figure 14 materials-14-06010-f014:**
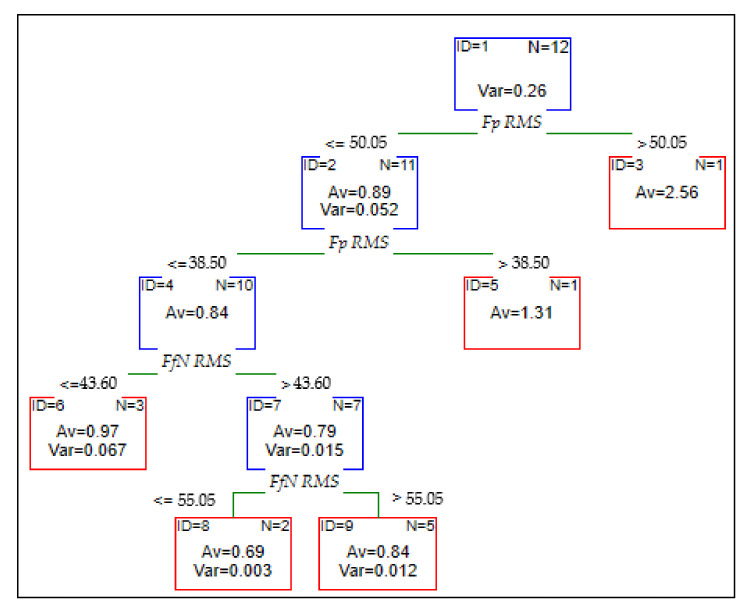
Structure of CART.

**Figure 15 materials-14-06010-f015:**
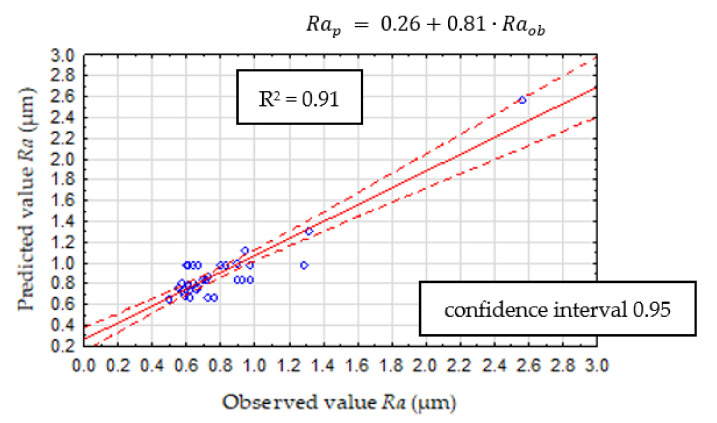
CART validation model.

**Figure 16 materials-14-06010-f016:**
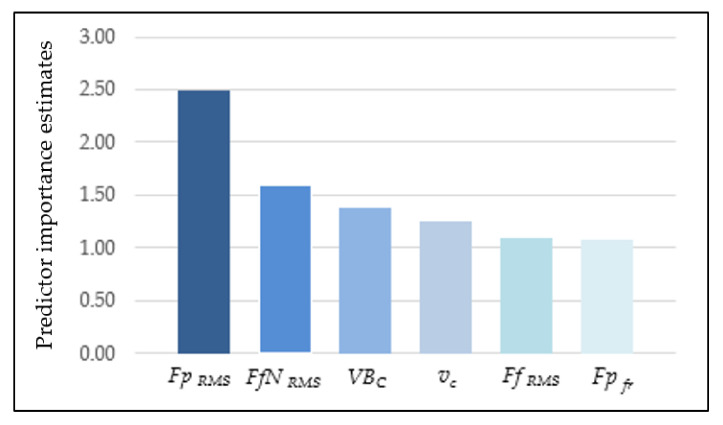
Validity of input parameters on surface roughness (ANN).

**Figure 17 materials-14-06010-f017:**
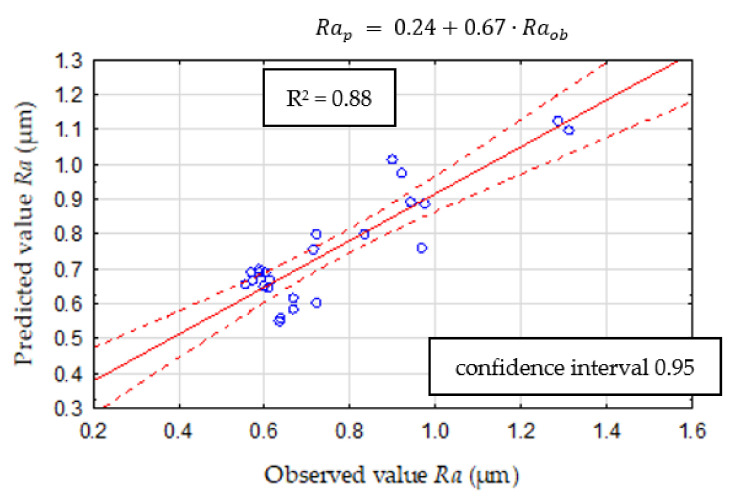
ANN validation model.

**Table 1 materials-14-06010-t001:** Typical physical properties of Duralcan^TM^ F3S.10S.

Density (g/cm^3^)	Electrical Conductivity (%IACS)	Specific Heat (cal/g·K)	Average Coefficient of Thermal Expansion (10^−6^/K)
2.71	34.2	0.21	20.7

**Table 2 materials-14-06010-t002:** Mechanical properties of F3S.10S composite.

Ultimate Strength (MPa)	Yield Strength (MPa)	Elongation (%)	Elastic Modulus (GPa)
221	165	2.6	98.6

**Table 3 materials-14-06010-t003:** Parameters of research plan.

Cutting Speed *v_c_* (m/min)	Spindle Speed *n* (rev/min)	Feed Per Tooth *f_z_* (mm/tooth)	Axial Infeed Depth *a_p_* (mm)	Radial Infeed Depth *a_e_* (mm)
300	9544	0.035	8	0.2
500	15,923			
900	28,662			

**Table 4 materials-14-06010-t004:** Structure of MLP model.

Educational Quality	Testing Quality	Validation Quality	Validation Error (Sum of Squares)	Activation Function in Hidden Layer	Activation Function in Output Layer
0.91	0.89	0.94	0.005	logistic	logistic

## Data Availability

The data presented in this study are available on request from the corresponding author.

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
