# Peer review of "Experimental Investigation of Surface Roughness in Milling of DuralcanTM Composite"

_materials, 2021, doi:10.3390/ma14206010_

Round 1
Reviewer 1 Report
In this paper, the analysis of surface roughness of aluminium matrix composites called Duralcan during finishing milling is described. The following surface quality parameters were investigated: arithmetic mean roughness and mean depth of roughness with respect to variable cutting speed. The paper describes the possibility of predicting surface roughness using machine learning algorithms. In order to find the best algorithm, classification and regression tree (CART) and artificial neural network (ANN) based pattern recognition models were compared. Based on the obtained models, the experiment proves the effectiveness of predicting the roughness based on the verification models.
The paper presents new information regarding the achievement of the desired surface quality after milling of Duralcan material. This information will certainly be of interest to readers dealing with machining.
Problematic parts of the text that need to be corrected are identified in the attached file.

Author Response
Dear Reviewer of Materials Journal,
Thank you for your comments and proposed changes. The changes have been made according to your suggestions.
Concerning individual items, I am sending the answers and corrections introduced to the article.
#Line 34 – double space has been removed
#Table 1 – thank you for your comment. The dot has been used.
#Figure 2 – descriptions of elements have been changed. I hope the figure is more readable now.
#Figures 3-5 – thank you for your comments. These figures have been edited.
#Figure 7 – the position of the dot has been changed.
#Line 186 – the size of the dot has been changed.
#Line 194 – the sentence has been moved.
#Line 198 – the size of the dot has been changed.
#Line 205 – the size of the dot has been changed.
#Figure 11 – thank you for your comments. This figure has been edited.
#Figure 12 – this figure has been edited.
#Figure 16 - this figure has been edited, and the dot has been added.
#References – the numbering of articles has been corrected, and the titles of articles have been verified.
Thank you very much for your consideration.
Sincerely,
Martyna Wiciak-Pikuła
Poznan University of Technology
3 Piotrowo St.
60-965 Poznan, POLAND
M: +48 790 412 919
E: martyna.r.wiciak@doctorate.put.poznan.pl

Reviewer 2 Report
Dear authors,
the paper is very interesting but there are some points to improve it.
1) Line 16. "Three roughness surface parameters..." Ra, Rz... What is the third parameter?
2) Introduction Line 92 "Results show..." On my mind, this part (lines 92-132) could be relocated to "Results and discussion" section of the paper.
3) Fig. 1 (microphoto): Black text "Spectrum 1" and "SiC" is hardly readable on the dark grey background. Please make it more readable, red or yellow for example.
4) Lines 149-152. Was some cutting fluid used in experiments or it was dry cutting? Was the same end mill used in all experiments? Please add the explanations because it can explain increasing of roughness with cutting time in a different way than was suggested in lines 177-178.
Author Response
Dear Reviewer of Materials Journal,
Thank you for your comments and proposed changes. The changes have been made according to your suggestions.
Concerning individual items, I am sending the answers and corrections introduced to the article.
#Line 16 – thank you for your comment. This is a mistake of surface roughness parameters numbers in the abstract. It should be two parameters. The number of these parameters has been made.
#Line 92 – thank you for this comment. Due to a literature review, this sentence and results are an example of the various experimental investigation of metal matrix composites milling (article number 18 in my references). I wanted to present multiple approaches to the optimization of machining conditions. In my opinion, this literature review is consistent and presents the metal matrix machining problem in various ways. Therefore, I am not sure that sentence (92-132) should be relocated to another chapter. Would you please think about it once again?
#Figure 1 – thank you for this comment. Descriptions of elements have been changed. I hope the figure is more readable now.
#Lines 149-152 – thank you for your comment. This experimental investigation was conducted without any fluid, and it was dry machining. The change has been made, and the information on dry machining in line 149 has been added. The one type of end mill was used in these studies. Three repetitions for each cutting speed were conducted, so generally, nine end mills ( 3 for 300 m/min, 3 for 500 m/min, and 3 for 900 m/min) have been used. Maybe it is not clear in the description of methods. The sentence has been changed in line 150, “a diamond-coated end mills were chosen to carry out research…” and the information has been added in line 152 “In Table 3, the research plan with one variable is presented”. I hope the description of the method is readable now.
Thank you very much for your consideration.
Sincerely,
Martyna Wiciak-Pikuła
Poznan University of Technology
3 Piotrowo St.
60-965 Poznan, POLAND
M: +48 790 412 919
E: martyna.r.wiciak@doctorate.put.poznan.pl

Reviewer 3 Report
The paper presents an interesting method of optimizing the surface roughness of hard to machine composite materials, like Duralcan.
I find the paper consistent, clear, well structured and and well presented. The only minor issue I found it was related to the References at the end. By mistake the numbers appears duplicate:
- 1.......
- 2. ....
- 3. ....
etc.
I would like to congratulate the authors of the paper for the results reached in their research.
Author Response
Dear Reviewer of Materials Journal,
Thank you for your comments and proposed changes. The changes have been made according to your suggestions.
Concerning individual items, I am sending the answers and corrections introduced to the article.
#References – the numbering of articles has been corrected.
Thank you very much for your positive opinion about the article.
Sincerely,
Martyna Wiciak-Pikuła
Poznan University of Technology
3 Piotrowo St.
60-965 Poznan, POLAND
M: +48 790 412 919
E: martyna.r.wiciak@doctorate.put.poznan.pl
